# NLRP3 Inflammasome Negatively Regulates RANKL-Induced Osteoclastogenesis of Mouse Bone Marrow Macrophages but Positively Regulates It in the Presence of Lipopolysaccharides

**DOI:** 10.3390/ijms23116096

**Published:** 2022-05-29

**Authors:** Mohammad Ibtehaz Alam, Megumi Mae, Fatima Farhana, Masayuki Oohira, Yasunori Yamashita, Yukio Ozaki, Eiko Sakai, Atsutoshi Yoshimura

**Affiliations:** 1Department of Periodontology and Endodontology, Nagasaki University Graduate School of Biomedical Sciences, 1-7-1 Sakamoto, Nagasaki 852-8588, Japan; bb55319201@ms.nagasaki-u.ac.jp (M.I.A.); m.mae@nagasaki-u.ac.jp (M.M.); bb55319202@ms.nagasaki-u.ac.jp (M.O.); yamachanpon@nagasaki-u.ac.jp (Y.Y.); ozaki@nagasaki-u.ac.jp (Y.O.); 2Department of Dental Pharmacology, Nagasaki University Graduate School of Biomedical Sciences, 1-7-1 Sakamoto, Nagasaki 852-8588, Japan; bb55319901@ms.nagasaki-u.ac.jp (F.F.); eiko-s@nagasaki-u.ac.jp (E.S.)

**Keywords:** NLRP3 inflammasome, interleukin-1β, bone remodeling, pyroptosis, reactive oxygen species, RANKL, osteoclast

## Abstract

In inflammatory bone diseases such as periodontitis, the nucleotide-binding oligomerization domain, leucine-rich repeat, and pyrin domain-containing 3 (NLRP3) inflammasome accelerates bone resorption by promoting proinflammatory cytokine IL-1β production. However, the role of the NLRP3 inflammasome in physiological bone remodeling remains unclear. Here, we investigated its role in osteoclastogenesis in the presence and absence of lipopolysaccharide (LPS), a Gram-negative bacterial component. When bone marrow macrophages (BMMs) were treated with receptor activator of nuclear factor-κB ligand (RANKL) in the presence of NLRP3 inflammasome inhibitors, osteoclast formation was promoted in the absence of LPS but attenuated in its presence. BMMs treated with RANKL and LPS produced IL-1β, and IL-1 receptor antagonist inhibited osteoclastogenesis, indicating IL-1β involvement. BMMs treated with RANKL alone produced no IL-1β but increased reactive oxygen species (ROS) production. A ROS inhibitor suppressed apoptosis-associated speck-like protein containing a caspase-1 recruitment domain (ASC) speck formation and NLRP3 inflammasome inhibitors abrogated cytotoxicity in BMMs treated with RANKL, indicating that RANKL induces pyroptotic cell death in BMMs by activating the NLRP3 inflammasome via ROS. This suggests that the NLRP3 inflammasome promotes osteoclastogenesis via IL-1β production under infectious conditions, but suppresses osteoclastogenesis by inducing pyroptosis in osteoclast precursors under physiological conditions.

## 1. Introduction

Bone is a calcified tissue that provides a framework for the body [1]. The integrity of bony structures is maintained by remodeling, whereby old or injured tissue is resorbed by osteoclasts and new bone is generated by osteoblasts [1,2]. However, under infectious conditions, excess osteoclast formation stemming from osteolytic diseases such as periodontitis and osteomyelitis induces high bone turnover and, ultimately, causes net bone loss and architectural bone decline [3,4]. Periodontitis in particular, which is associated with inflammatory alveolar bone loss, is a health threat at all ages [5]. Pathogenic invasion from the oral environment can activate inflammation, which increases osteoclast activity in the alveolar bone and disrupts the balance between the activity of osteoclasts and osteoblasts [6,7]. It is thus conceivable that the external environment plays a crucial role in regulating bone remodeling.

Periodontitis is commonly initiated by accumulated dental deposits such as dental plaque and calculus [5,8]. In response to these deposits, inflammatory mediators such as prostaglandins, matrix metalloproteinases, and cytokines, are released in periodontal tissue [5,9]. Among them, proinflammatory cytokine IL-1β is a potent activator of bone resorption [10]. Bacterial components contained in dental deposits can be recognized by host immune sensors such as Toll-like receptors (TLRs), leading to the nuclear translocation of nuclear factor (NF)-κB, resulting in pro-IL-1β production, which remains biologically inactive in the cytosol [6,9,11]. For IL-1β maturation, 31 kDa pro-IL-1β must be proteolytically processed to generate the active 17 kDa form [12]. During this process, bacterial components or dental calculus activate the nucleotide-binding oligomerization domain, leucine-rich repeat, and pyrin domain-containing 3 (NLRP3) inflammasome, which plays an important role in the immune system and consists of NLRP3, apoptosis-associated speck-like protein containing a caspase-1 recruitment domain (ASC), and pro-caspase-1 [8,13,14]. During formation of the NLRP3 assembly, cysteine protease pro-caspase-1 converts into active caspase-1 via autocatalysis, promoting IL-1β maturation [15,16]. Mature IL-1β has the capacity to up-regulate osteoclast formation and promote bone resorption in inflammatory diseases such as periodontitis and osteomyelitis [4,5]. Although the balance between bone formation and resorption is maintained under healthy conditions, the NLRP3 inflammasome plays an important role in IL-1β production, which under infectious conditions shifts the balance of bone homeostasis toward greater bone resorption.

It is widely recognized that the NLRP3 inflammasome is involved in bone resorption following infection challenge [17,18]. However, the role of the NLRP3 inflammasome under physiological conditions remains unclear. A recent study revealed that bone matrix components activate the NLRP3 inflammasome and that bone particles cause exuberant osteoclastogenesis in wild-type (WT) cells, but less so in *Nlrp3*^−/−^ cells in the presence of receptor activator of nuclear factor-κB ligand (RANKL), a master regulator of osteoclast formation. This suggests that the NLRP3 inflammasome activation by bone matrix components, which can be generated in both physiological and pathological bone turnover, amplifies bone resorption [19]. Under physiological conditions, bone resorption by osteoclasts and bone formation by osteoblasts are repeated continuously to maintain bone homeostasis [1,2]. During osteoclastogenic events, RANKL interacts with receptor activator of nuclear factor-κB (RANK) and mediates osteoclast formation. RANKL–RANK interaction can induce cellular stress, which markedly elevates levels of reactive oxygen species (ROS) [20,21]. ROS are thought to be one of the intracellular messengers responsible for NLRP3 inflammasome activation [22]. Considering these interconnected links between RANKL, ROS, and NLRP3, we hypothesized that RANKL may activate the NLRP3 inflammasome in osteoclastogenesis under physiological conditions. We therefore investigated the role of the NLRP3 inflammasome in osteoclast differentiation in bone marrow macrophages (BMMs) under infectious and physiological conditions. In addition, we elucidated the role of RANKL in NLRP3 inflammasome activation during osteoclastogenesis.

For this study, we generated osteoclasts from BMMs under the presence or absence of NLRP3 inflammasome inhibitors to investigate the role of the NLRP3 inflammasome in osteoclastogenesis under physiological and infectious conditions. We used pit assays to assess the bone resorption capacity. Inflammasome assembly was confirmed via ASC speck formation in the presence or absence of a ROS inhibitor. IL-1β levels were analyzed via quantitative reverse-transcription–polymerase chain reaction (qRT-PCR) and enzyme-linked immunosorbent assay (ELISA). Finally, we performed cytotoxicity assays to further analyze the underlying mechanisms.

## 2. Results

### 2.1. Role of NLRP3 Inflammasome and IL-1β in Osteoclastogenesis in the Presence and Absence of Lipopolysaccharide (LPS)

To investigate whether the NLRP3 inflammasome regulates RANKL-induced osteoclast formation, we measured the gene expression of NLRP3 inflammasome components and IL-1β in BMMs treated with RANKL in the presence or absence of LPS. RANKL induced osteoclastogenesis in a dose-dependent manner in the presence of macrophage colony-stimulating factor (M-CSF; Figure 1a). To understand the role of the NLRP3 inflammasome under infectious conditions, we treated BMMs with RANKL (5 ng/mL) and increasing concentrations of bacterial LPS (0, 1, 10, and 100 ng/mL). LPS stimulation increased osteoclast abundance (Figure 1b). RANKL (20 ng/mL) alone significantly upregulated mRNA expression of *Nlrp3* and *Caspase-1*, but not of *Asc* or *Il-1β* (Figure 1c). RANKL also significantly upregulated protein expression of NLRP3 and cleaved caspase-1, suggesting the involvement of NLRP3 inflammasome in RANKL-induced osteoclastogenesis (Figure 1d). When BMMs were treated with RANKL and LPS, gene expression of *Nlrp3*, *Caspase-1*, and *Il-1β* was upregulated, but that of *Asc* was downregulated. These findings clearly demonstrate that RANKL can upregulate *Nlrp3* and *Caspase-1* but not *Il-1β* in the absence of LPS.

We then analyzed IL-1β production in BMMs. When BMMs were treated with RANKL, IL-1β was not detected in either the supernatant or the lysate (Figure 2a). When the cells were treated with RANKL in the presence of LPS, however, IL-1β was detected in the lysate but not in the supernatant (Figure 2b). To determine whether IL-1β is involved in osteoclastogenesis, we treated BMMs with RANKL and LPS in the presence or absence of recombinant IL-1 receptor antagonist (rIL-1ra). rIL-1ra did not affect osteoclast formation in the absence of LPS (Figure 2c), but it inhibited osteoclast formation in its presence (Figure 2d). These results indicate that IL-1β is involved in osteoclast formation in BMMs treated with RANKL in the presence of LPS, but not in its absence.

### 2.2. Differential Effects of NLRP3 Inflammasome Inhibitors on Osteoclast Formation

We analyzed the role of the NLRP3 inflammasome on osteoclast differentiation using MCC950, a specific NLRP3 inflammasome inhibitor, and Z-YVAD-FMK, a caspase-1 inhibitor. Both inhibitors significantly increased the number of osteoclasts in BMMs treated with RANKL (Figure 3a,b). However, they reduced osteoclast abundance in BMMs treated with RANKL and LPS (Figure 3c,d). These findings indicate that the NLRP3 inflammasome affects RANKL-mediated osteoclastogenesis negatively in the absence of LPS but positively in its presence. The osteoclastogenic effects of these specific inhibitors were confirmed by the increase in mRNA expression of the osteoclastogenesis markers, nuclear factor of activated T-cells 1 (*Nfatc1*), cathepsin K (*Ctpk*), and osteoclast-associated receptor (*Oscar*) (Figure 3e).

To confirm the role of NLRP3 inflammasome inhibitors, we performed a pit assay of RANKL-mediated osteoclastogenesis. RANKL-treated BMMs showed only mild resorbing activity, but MCC950 and Z-YVAD-FMK increased osteoclast formation in RANKL-treated BMMs, which ultimately generated a greater number of pit areas (Figure 4a,b). Bone resorption was then analyzed via fluorescent-substrate release in BMM culture supernatants. Similarly to the pit assay, MCC950 and Z-YVAD-FMK increased fluorescence intensity, confirming the negative role of the NLRP3 inflammasome in RANKL-mediated osteoclastogenesis in the absence of LPS (Figure 4c).

### 2.3. NLRP3 Inflammasome Negatively Regulates Osteoclastogenesis by Inducing Pyroptotic Cell Death

ROS are important intracellular messengers that can be produced by RANKL during osteoclastogenesis [21]. We examined ROS levels in RANKL-treated BMMs. When the BMMs were exposed to higher concentrations of RANKL, intracellular ROS production was upregulated (Figure 5a,b). However, RANKL failed to induce ROS production in the presence of YCG063, a specific ROS inhibitor.

Because ROS are thought to activate the NLRP3 inflammasome [22], we speculated that RANKL may activate the NLRP3 inflammasome via ROS production. When we treated BMMs with RANKL, we observed ASC speck formation (Figure 6). However, this effect was abrogated by YCG063, suggesting that the induction of ROS production by RANKL plays an important role in activating the NLRP3 inflammasome.

RANKL therefore activated the NLRP3 inflammasome, and NLRP3 inflammasome inhibitors boosted the osteoclast differentiation induced by RANKL, revealing the negative role of the NLRP3 inflammasome in RANKL-mediated osteoclastogenesis. Since NLRP3 activation facilitates gasdermin D maturation, which mediates pyroptosis [23], we hypothesized that RANKL would induce pyroptotic cell death in BMMs. To test this hypothesis, we performed cytotoxicity assays using BMMs treated with RANKL in the presence or absence of MCC950 or Z-YVAD-FMK. Both MCC950 and Z-YVAD-FMK reduced the cytotoxic effects of RANKL, indicating that RANKL does induce pyroptotic cell death in BMMS (Figure 7a). The cleaved gasdermin D was detected in RANKL-treated BMMS, but not in untreated cells, demonstrating that RANKL induced gasdermin D maturation. (Figure 7b). When BMMs were treated with RANKL in the presence of LPS, Z-YVAD-FMK partially decreased the cytotoxicity of RANKL but MCC950 did not, revealing that RANKL exhibits lower cytotoxicity to BMMs in the presence of LPS (Figure 7c).

## 3. Discussion

In this study, we demonstrated how the NLRP3 inflammasome is involved in osteoclast formation under various conditions. The abundance of osteoclasts increased in the presence of NLRP3 inflammasome inhibitors when BMMS were treated with RANKL in the absence of LPS. Moreover, these inhibitors generated more pit areas and increased the fluorescence intensity of bone substrates, indicating that the NLRP3 inflammasome negatively regulates osteoclast formation. However, osteoclast formation was suppressed in the presence of NLRP3 inflammasome inhibitors in BMMs treated with RANKL in the presence of LPS, suggesting that NLRP3 accelerates osteoclast formation. These results suggest that the NLRP3 inflammasome regulates osteoclast formation positively under infectious conditions but negatively under physiological conditions.

In response to RANKL and LPS treatment, mRNA levels of *Nlrp3*, *Caspase-1*, and (more importantly) *Il-1β* were upregulated. Although we could not detect IL-1β in the culture supernatant of BMMs treated with RANKL and LPS, we did detect it in the cell lysate. IL-1β may be consumed in the culture supernatant of BMMs during differentiation to osteoclasts. Further, rIL-1ra suppressed osteoclast formation from BMMs treated with RANKL and LPS. These results indicate that IL-1β is involved in the osteoclastogenesis induced by RANKL in the presence of LPS. This is consistent with previous studies showing that IL-1β accelerates osteoclast formation in inflammatory diseases such as periodontitis and osteomyelitis [4,5]. These findings suggest that the NLRP3 inflammasome contributes positively to osteoclastogenesis under infectious conditions, via IL-1β production (Figure 8).

In the absence of LPS, RANKL upregulated the expression of *Nlrp3* and *Caspase-1*, but not *Il-1β*. We detected no IL-1β production in either the BMM culture supernatant or the cell lysate. Further, rIL-1ra did not inhibit the osteoclastogenesis of BMMs treated with RANKL. Combined, these results indicate that IL-1β was not involved in the osteoclastogenesis that was induced by RANKL alone. However, the number of osteoclasts induced by RANKL increased in the presence of NLRP3 inflammasome inhibitors, revealing the negative role played by the NLRP3 inflammasome in osteoclastogenesis. As previously shown, ROS levels increase in BMMs treated with RANKL [20,21], and ROS are key activators of the NLRP3 inflammasome [22]. Further, a ROS inhibitor suppressed ASC speck formation in BMMs treated with RANKL. These findings indicate that RANKL activates the NLRP3 inflammasome via ROS production. Assembly of the NLRP3 inflammasome results in the processing of caspase-1, which cleaves gasdermin D to its active form to induce pyroptosis [23]. Inhibitors of the NLRP3 inflammasome suppressed the cell death of BMMs stimulated with RANKL, indicating that RANKL induces pyroptotic cell death via ROS and the NLRP3 inflammasome in BMMs. Based on these results, RANKL may negatively regulate osteoclastogenesis by inducing pyroptotic cell death (Figure 8). To the best of our knowledge, this is the first evidence uncovering the mechanism by which the NLRP3 inflammasome regulates RANKL-induced osteoclastogenesis.

This study has demonstrated that the NLRP3 inflammasome can regulate osteoclast formation positively, and possibly also negatively, in BMMs, depending on the environment. Under infectious conditions, the NLRP3 inflammasome disrupts bone homeostasis by promoting osteoclast formation via IL-1β production, possibly to metabolize injured bone tissue and facilitate pathogen elimination around the bone. However, under physiological conditions, the NLRP3 inflammasome suppresses osteoclast formation via pyroptosis. Here, the NLRP3 inflammasome probably plays a crucial role in protecting bone from osteoclast overactivity, and thus helps to maintain bone homeostasis. In addition, this study uncovered the role of the NLRP3 inflammasome in RANKL-mediated physiological osteoclast differentiation. It is still necessary to clarify whether the NLRP3 inflammasome protects against excessive bone loss, using animal models. Future studies will elucidate the exact role of the NLRP3 inflammasome in regulating bone remodeling in infectious, as well as inflammatory, bone diseases such as rheumatoid arthritis and osteoporosis.

## 4. Materials and Methods

### 4.1. Reagents

Minimum Essential Media alpha (MEMα) and Dulbecco’s phosphate-buffered saline (D-PBS) were purchased from Fujifilm Wako Pure Chemical (Osaka, Japan). Fetal bovine serum (FBS) was purchased from Hyclone (Logan, UT, USA). Penicillin-streptomycin, anti-rabbit secondary antibody conjugated with Alexa Fluor 488 (A-11008) and CellROX Deep Red Reagent were purchased from Thermo Fisher Scientific (Waltham, MA, USA). ELISA kits for mouse IL-1β (DuoSet), recombinant mouse M-CSF, and recombinant mouse RANKL were purchased from R&D Systems (Minneapolis, MN, USA). Ultra-pure LPS from *Escherichia coli* O111:B4 was obtained from InvivoGen (San Diego, CA, USA). Erythrocyte lysis buffer BD Pharm Lyse were purchased from BD Biosciences (San Jose, CA, USA). Dimethyl sulfoxide (DMSO) and tartrate-resistant acid phosphatase (TRAP) staining kits were purchased from Sigma–Aldrich (St. Louis, MO, USA). NLRP3 inflammasome inhibitor MCC950 was purchased from Cayman Chemical (Ann Arbor, MI, USA) and caspase-1 inhibitor Z-YVAD-FMK was obtained from Calbiochem-EMD Millipore (Darmstadt, Germany). The Bone Resorption Assay Kit 48 was purchased from Iwai Chemicals Company (Tokyo, Japan). Mouse rIL-1ra was purchased from Biovision (Milpitas, CA, USA). Takara SYBR Premix Ex Taq (Tli RNase H Plus) was obtained from Takara Bio (Otsu, Japan). A rabbit anti-ASC antibody (AL177) was purchased from AdipoGen (San Diego, CA, USA) and CytoTox 96 nonradioactive assay kits were purchased from Promega (Madison, WI, USA). Antibodies for Western blot analysis, anti-NLRP3, anti-gasdermin D, anti-pro caspase-1, anti-caspase-1 rabbit monoclonal antibodies and horseradish peroxidase-conjugated secondary antibodies, were purchased from Cell Signaling Technology (Danvers, MA, USA).

### 4.2. BMM Isolation

Bone marrow (BM) cells were eluted from the femur and tibiae of six-week-old male BALB/c mice, as previously described [24]. Briefly, BM was flushed from bones using MEMα, and BM cells were collected via centrifugation, treated with erythrocyte lysis buffer for 10 min at room temperature, and washed with D-PBS. The BM cells were then incubated in MEMα supplemented with 10% FBS, 100 U/mL penicillin, and 100 μg/mL streptomycin in the presence of M-CSF (5 ng/mL) overnight. Nonadherent BM cells were then collected and incubated in 10 cm dishes for 48 h in MEMα containing M-CSF (30 ng/mL), 10% FBS, 100 U/mL penicillin, and 100 μg/mL streptomycin. Cells attached to the dishes after 48 h were collected and used as BMMs. Cells were incubated at 37 °C and in a 5% CO_2_ atmosphere.

### 4.3. In Vitro Osteoclastogenesis

To induce osteoclast differentiation, BMMs (1 × 10^4^ cells/well) were seeded in 96-well plates, incubated with M-CSF (30 ng/mL) and RANKL (0, 2.5, 5, 10, or 20 ng/mL) in MEMα supplemented with 10% FBS, 100 U/mL penicillin, and 100 μg/mL streptomycin for 72 h. For the inhibition assays, BMMs were incubated with M-CSF (30 ng/mL) and RANKL (10 ng/mL) in the presence or absence of rIL-1ra, MCC950, or Z-YVAD-FMK in the same concentration of M-CSF for 72 h.

Alternatively, BMMs (1 × 10^4^ cells/well) were seeded in 96-well plates, incubated with M-CSF (30 ng/mL) and RANKL (5 ng/mL) in MEMα supplemented with 10% FBS, 100 U/mL penicillin, and 100 μg/mL streptomycin for 48 h. The cells were then stimulated with ultrapure LPS (0, 1, 10, or 100 ng/mL) in the presence of the same concentrations of RANKL and M-CSF for another 48 h. For the inhibition assays, the cells were incubated with the same concentrations of M-CSF, RANKL, and LPS in the presence or absence of rIL-1ra, MCC950, or Z-YVAD-FMK for the final 48 h.

Cells in 96-well plates were fixed in 4% paraformaldehyde (PFA) for 15 min at 4 °C, treated with 0.2% Triton X-100 in PBS at room temperature for approximately 8–10 min, treated with TRAP staining solution for 10 min at 37 °C, and rinsed three times with distilled water. TRAP-positive cells with three or more nuclei were counted as osteoclasts.

### 4.4. IL-1β Concentration Measurements

BMMs (5 × 10^4^ cells/well) were seeded in 24-well plates, incubated with M-CSF (30 ng/mL), and RANKL (0, 2.5, 5, 10, or 20 ng/mL) for 7 h. Alternatively, BMMs (5 × 10^4^ cells/well) were seeded in 24-well plates, then incubated with M-CSF (30 ng/mL) and RANKL (5 ng/mL) for 48 h. The cells were then stimulated with LPS (0, 1, 10, or 100 ng/mL) for 7 h in combination with the same concentrations of M-CSF and RANKL. For positive controls, cells were primed with LPS (1 μg/mL) for 3 h, then stimulated with adenosine triphosphate (ATP; 4 mM) for another 45 min. The culture supernatant was collected after centrifugation. The cell lysate was extracted using ice-cold 1% Triton X-100 diluted in D-PBS for 30 min at 4 °C and collected after centrifugation. The concentration of IL-1β was then measured via ELISA, according to the manufacturer’s protocol.

### 4.5. qRT-PCR Analysis

To analyze the mRNA levels of inflammasome-related genes such as *Nlrp3*, *Caspase-1*, *Asc*, and *Il-1β,* BMMs (1 × 10^6^ cells/dish) were incubated with M-CSF (30 ng/mL) in 60 cm dishes. Following overnight incubation, the cells were treated with or without RANKL (20 ng/mL) in the presence or absence of YCG063 (20 μM) for 7 h. To analyze the mRNA levels of osteoclast-related genes such as *Nfatc1*, *Ctpk* and *Oscar*, BMMs (1 × 10^6^ cells/dish) were incubated with M-CSF (30 ng/mL) and RANKL (20 ng/mL) in the presence or absence of MCC950 or Z-YVAD-FMK for 60 h. Total RNA was extracted from cells using a NucleoSpin RNA purification kit (Takara Bio, Otsu, Japan) with on-column DNase treatment, according to the manufacturer’s instructions. For each sample, 2 μg of total RNA was converted to first-strand cDNA using avian myeloblastosis virus reverse transcriptase (Promega, Madison, WI, USA) at 25 °C for 10 min, followed by 50 min at 42 °C, then 15 min at 70 °C using a Takara PCR thermal cycler (Takara Bio, Otsu, Japan). The cDNA was treated with RNase H and purified using the QIAprep Spin Miniprep Kit (Qiagen, Hilden, Germany). Primer sequences used were designed using the Primer3 Input (v0.4.0) software and obtained from Hokkaido System Science (Sapporo, Japan). Primer sequences were as follows: *Nlrp3* forward, 5′-ATGCTGCTTCGACATCTCCT-3′ and reverse, 5′-AACCAATGCGAGATCCTGAC-3′; *Caspase-1* forward, 5′-CACAGCTCTGGAGATGGTGA-3′ and reverse, 5′-TCTTTCAAGCTTGGGCACTT-3′; *Asc* forward, 5′-TCACAGAAGTGGACGGAGTG-3′ and reverse, 5′-CTCCAGGTCCATCACCAAGT-3′; *Il-1β* forward, 5′-AACCTGCTGGTGTGTGACGTTC-3′ and reverse, 5′-CAGCACGAGGCTTTTTTGTTGT-3′; *Nfatc1* forward, 5′-GGGTCAGTGTGACCGAAGAT-3′ and reverse, 5′-GGAAGTCAGAAGTGGGTGGA-3′; *Ctpk* forward, 5′-CCAGTGGGAGCTATGGAAGA-3′ and reverse, 5′-AAGTGGTTCATGGCCAGTTC-3′; *Oscar* forward, 5′-TGCTGTGCCAATCACAAGTA-3′ and reverse, 5′-AGGGAAACCTCATCCGTTT-3′; *GAPDH* forward, 5′-GGAGGAACCTGCCAAGTATG-3′, and reverse, 5′-TGGGAGTTGCTGTTGAAGTC-3′. Comparative quantification for *Nlrp3*, *Caspase-1*, *Asc*, and *Il-1β* was completed with SYBR Premix Ex Taq using the Mx3000 P qPCR System (Agilent Technologies, Santa Clara, CA, USA). Amplification conditions were as follows: 95 °C for 10 s, followed by 40 cycles of 95 °C for 5 s, 54 °C (*Oscar*), 56 °C *(Nlrp3* and *Caspase-1*) or 58 °C (*Il-1β*, *Asc*, *Nfatc1* and *Ctpk*) for 20 s, and a final cycle of 95 °C for 1 min, 55 °C for 30 s, and 95 °C for 30 s. A melting curve analysis was used to confirm that the proper PCR products had been amplified in all samples. The relative expression ratio of *Nlrp3*, *Caspase-1*, *Asc*, and *Il-1β* mRNA was calculated based on PCR efficiency and threshold cycle differences between test samples (stimulated cells) and a calibrator (unstimulated cells). Target gene expression was normalized using *GAPDH* gene expression. Calibrator mRNA levels were set to 1.

### 4.6. ASC Speck Formation

BMMs (5 × 10^4^ cells/well) were incubated with M-CSF (30 ng/mL) in 96-well plates overnight, and then treated with RANKL (0 or 20 ng/mL) in the presence or absence of YCG063 (20 μM) for 24 h. For positive controls, cells were primed with LPS (1 μg/mL) for 3 h and further stimulated with ATP (5 mM) for 90 min, then fixed with 4% PFA in PBS for 30 min at 4 °C and permeabilized with 0.2% Triton X-100 in PBS at room temperature for 10 min. A blocking solution containing 0.2% gelatin was applied for 60 min at room temperature. The cells were then incubated with a rabbit anti-ASC antibody (1:250 dilution) overnight at 4 °C. After the cells had been washed with blocking buffer thrice, goat anti-rabbit secondary antibody conjugated with Alexa Fluor 488 (1:500 dilution) was applied for 60 min. The cells were then counterstained with Hoechst 33342 for 30 min. Finally, ASC speck formation was analyzed via fluorescent microscopy (BZ-X800, All-in-One Fluorescence Microscope, Keyence, Osaka, Japan).

### 4.7. Cytotoxicity Assay

BMMs (1 × 10^4^ cells/well) were seeded in 96-well plates and incubated with M-CSF (30 ng/mL), RANKL (0 or 10 ng/mL), and 4% FBS in the presence or absence of MCC950 or Z-YVAD-FMK for 72 h. Alternatively, BMMs (1 × 10^4^ cells/well) were seeded in 96-well plates and incubated with M-CSF (30 ng/mL), RANKL (0 or 5 ng/mL), and 10% FBS for 48 h. The cells were then stimulated with LPS (10 ng/mL) with 1% FBS and the same concentrations of M-CSF and RANKL, in the presence or absence of MCC950 or Z-YVAD-FMK, for 48 h. As a negative control, cells were treated with 30 ng/mL M-CSF alone. As a positive control, cells were frozen (−80 °C) and thawed for two cycles. After stimulation, the plates were centrifuged and the cell culture supernatant was harvested. Lactate dehydrogenase (LDH) in the supernatant was then measured using a CytoTox 96 nonradioactive assay, according to the manufacturer’s protocol. Cytotoxicity percentage was determined based on the following calculation [25]:(1)$$\mathrm{Cytotoxicity}\;(\%)=\frac{\mathrm{Experimental}\;\mathrm{LDH}\;\mathrm{release}\;\left(\mathrm{OD}\right)-\mathrm{Background}\;\mathrm{LDH}\;\mathrm{release}\;\left(\mathrm{OD}\right)}{\mathrm{Maximum}\;\mathrm{LDH}\;\mathrm{release}\;\left(\mathrm{OD}\right)-\mathrm{Background}\;\mathrm{LDH}\;\mathrm{release}\;\left(\mathrm{OD}\right)}\times 100$$

### 4.8. Bone-Resorption Assay

BMMs (5 × 10^4^ cells/well) were seeded in 48-well plates and cultured in MEMα (without phenol red) supplemented with 10% FBS, 100 U/mL penicillin, 100 µg/mL streptomycin containing M-CSF (30 ng/mL), and RANKL (20 ng/mL), in the presence or absence of MCC950 and Z-YVAD-FMK, for 4 d. The medium was then replaced with fresh medium containing the same concentrations of these reagents and incubated for another 6 d. After incubation, a bone-resorption pit assay was performed, according to the manufacturer’s instructions. Briefly, culture supernatant was transferred to a black plate. Bone-resorption assay buffer was added and fluorescence was measured using a fluorescent plate reader (BMG FLUOstar OPTIMA, BMG LABTECH, Ortenberg, Germany). After the supernatant had been removed, the calcium-coated plates were treated with 5% sodium hypochlorite to remove the cells. Images of the pit areas were then captured using a microscope and analyzed using ImageJ software (v1.52a; http://imagej.nih.gov/ij/, accessed on 28 March 2022) [24].

### 4.9. ROS Detection Assay

To detect intracellular ROS production, BMMs (5 × 10^4^ cells/well) were seeded in 96-well plates and cultured overnight with M-CSF (30 ng/mL). The cells were then treated with RANKL (0, 2.5, 5, 10, or 20 ng/mL) in the presence of CellROX Deep Red Reagent, at a final concentration of 5 μM, for 30 min. For inhibition assays, BMMs were pretreated with YCG063 (20 μM) for 1 h. The cells were then counterstained with Hoechst 33342 for 15 min, washed with PBS, and the fluorescence was measured using a fluorescent plate reader (BMG FLUOstar OPTIMA). ROS-expressing cells were also analyzed via fluorescence microscopy (BZ-X800, All-in-One Fluorescence Microscope).

### 4.10. Western Blot Analysis

BMMs (5 × 10^6^ cells/dish) were treated with RANKL (0, 20 ng/mL) and M-CSF (30 ng/mL) for 24 h and subjected to Western blot analysis as previously described [26]. Briefly, cells were rinsed twice with ice-cold PBS, and lysed in a cell lysis buffer (50 mM Tris-HCl (pH 8.0), 1% Nonidet P-40, 0.5% sodium deoxycholate, 0.1% sodium dodecyl sulfate (SDS), 150 mM NaCl, 1 mM PMSF, and proteinase inhibitor cocktail). An equal amount of protein (5 µg) was applied to each lane of SDS-polyacrylamide gel electrophoresis, followed by transfer onto a polyvinylidene difluoride membrane. The blots were blocked with 3% skim milk in Tris-buffered saline for 1 h at 25 °C. Blots were then incubated with anti-NLRP3, anti-pro Caspase-1, anti-cleaved caspase-1, anti-gasdermin D, or anti-GAPDH primary antibodies overnight at 4 °C, washed, incubated with horseradish peroxidase-conjugated secondary antibodies, and finally detected with Immobilon Forte Western HRP substrate (Merck, Darmstadt, Germany). The immunoreactive bands were analyzed using a LAS-4000mini (Fujifilm, Tokyo, Japan) and relative level of proteins were determined using ImageJ software.

### 4.11. Statistical Analysis

Statistical differences among groups were assessed using a one-factor analysis of variance (ANOVA) with Tukey–Kramer tests. Differences between two groups were analyzed using *t*-tests. All data were analyzed using Stat Mate V (ATMS, Tokyo, Japan).

## Figures and Tables

**Figure 1 ijms-23-06096-f001:**
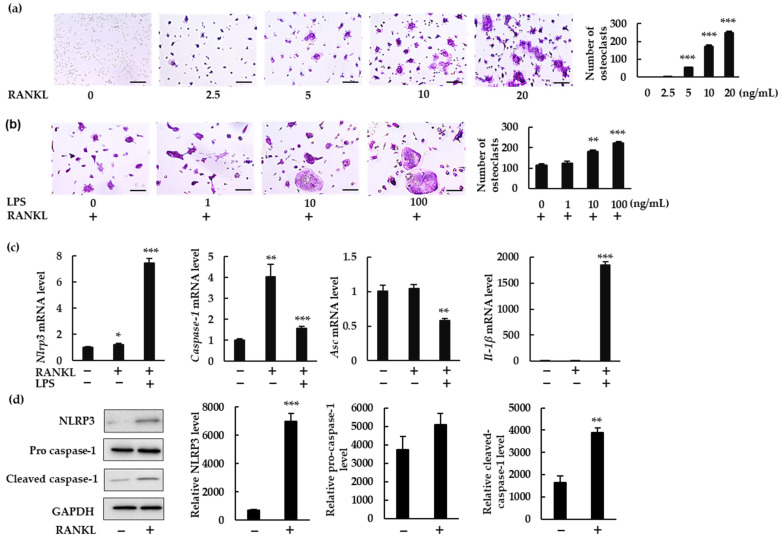
NLRP3 inflammasome-related gene and protein expression in bone marrow macrophages (BMMs) treated with receptor activator of nuclear factor-κB ligand (RANKL) in the presence or absence of lipopolysaccharide (LPS). (**a**) BMMs were treated with macrophage-colony-stimulating factor (M-CSF; 30 ng/mL) and RANKL (0, 2.5, 5, 10, or 20 ng/mL) for 72 h. (**b**) BMMs were treated with RANKL (5 ng/mL) and M-CSF (30 ng/mL) for 48 h, then stimulated with LPS (0, 1, 10, or 100 ng/mL) for an additional 48 h in the presence of the same concentrations of M-CSF and RANKL. (**a**,**b**) Cells were then subjected to tartrate-resistant acid phosphatase (TRAP) staining. TRAP-positive cells with three or more nuclei were counted as osteoclasts. (**c**) BMMs were treated with RANKL (0 or 20 ng/mL) and M-CSF (30 ng/mL) in the presence or absence of LPS (10 ng/mL) for 7 h. Total RNA was extracted from cells and the relative expression of *Nlrp3*, *Caspase-1*, *Asc*, and *Il-1β* mRNA was determined using quantitative reverse-transcription–polymerase chain reaction (qRT-PCR). (**d**) BMMs were treated with RANKL (0 or 20 ng/mL) and M-CSF (30 ng/mL) for 24 h. Cell lysates were then prepared and subjected to sodium dodecyl sulfate polyacrylamide gel electrophoresis (SDS-PAGE) followed by Western blotting using anti-NLRP3, anti-pro caspase-1, anti-cleaved caspase-1 and anti-glyceraldehyde-3-phosphate dehydrogenase (GAPDH) antibodies. Relative levels of protein bands were determined using ImageJ software. Scale bars: 150 μm. Results are expressed as the means ± standard error (SE) of triplicate assays and the images are representative of three independent experiments. * *p* < 0.05, ** *p* < 0.01, *** *p* < 0.001 compared with the control.

**Figure 2 ijms-23-06096-f002:**
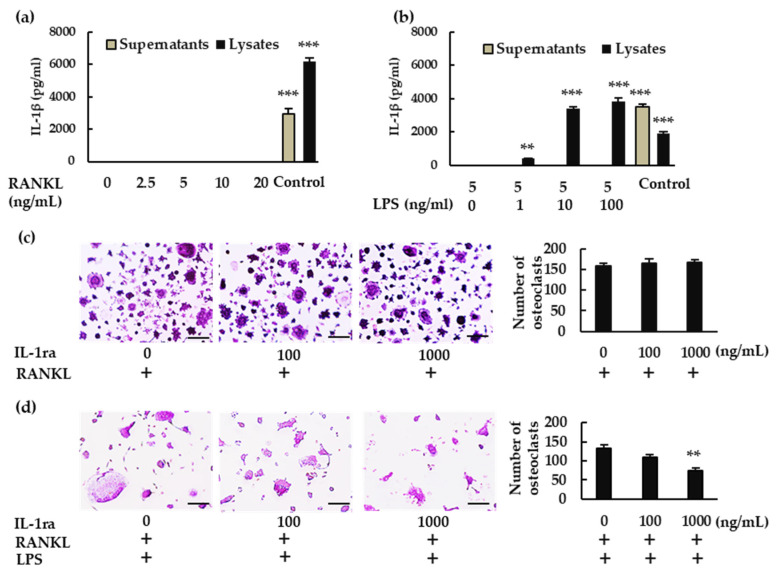
Production of IL-1β and its involvement in osteoclastogenesis. BMMS were treated with (**a**) M-CSF (30 ng/mL) and RANKL (0, 2.5, 5, 10, or 20 ng/mL) for 7 h, or with (**b**) RANKL (5 ng/mL) and M-CSF (30 ng/mL) for 48 h, then stimulated with LPS (0, 1, 10, or 100 ng/mL) for another 7 h in the presence of the same concentrations of M-CSF and RANKL. For positive controls, cells were primed with LPS (1 μg/mL) for 3 h, then stimulated with adenosine triphosphate (ATP; 4 mM) for another 45 min. (**a**,**b**) IL-1β production was measured via enzyme-linked immunosorbent assay (ELISA). (**c**) BMMs were treated with M-CSF (30 ng/mL) and RANKL (10 ng/mL) for 72 h in the presence of the indicated concentrations of rIL-1ra. (**d**) BMMs were treated with RANKL (5 ng/mL) and M-CSF (30 ng/mL) for 48 h, then stimulated with LPS (10 ng/mL) in the presence of the indicated concentrations of rIL-1ra for another 48 h. (**c**,**d**) The cells were then subjected to TRAP staining, and TRAP-positive cells with three or more nuclei were counted as osteoclasts. Scale bars: 150 μm. Results are expressed as the means ± SE of triplicate assays and the images are representative of three independent experiments. ** *p* < 0.01, *** *p* < 0.001 compared with the control.

**Figure 3 ijms-23-06096-f003:**
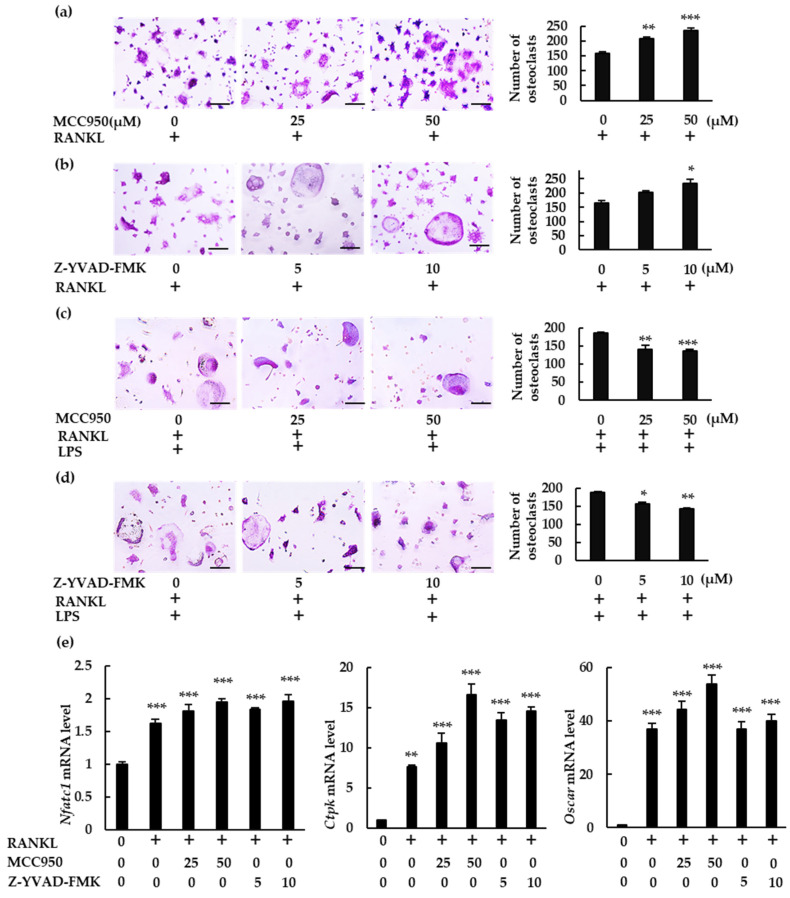
Differential effects of NLRP3 inflammasome inhibitors on osteoclast formation. (**a**,**b**) BMMs were treated with M-CSF (30 ng/mL) and RANKL (10 ng/mL) for 72 h in the presence of the indicated concentrations of MCC950 (an NLRP3 inhibitor) or Z-YVAD-FMK (a caspase-1 inhibitor). (**c**,**d**) BMMs were treated with RANKL (5 ng/mL) and M-CSF (30 ng/mL) for 48 h, then treated with LPS (10 ng/mL) and the same concentrations of M-CSF and RANKL for an additional 48 h in the presence of the indicated concentrations of MCC950 or Z-YVAD-FMK. Cells were subjected to TRAP staining, and TRAP-positive cells with three or more nuclei were counted as osteoclasts. (**e**) BMMs were treated with RANKL (0 or 20 ng/mL) and M-CSF (30 ng/mL) in the presence or absence of MCC950 or Z-YVAD-FMK for 60 h. Total RNA was extracted from cells and the relative expression of *Nfatc1*, *CtpK*, and *Oscar* mRNA was determined using qRT-PCR. Scale bars: 150 μm. Results are expressed as the means ± SE of triplicate assays and the images are representative of three independent experiments. * *p* < 0.05, ** *p* < 0.01, *** *p* < 0.001 compared with the control.

**Figure 4 ijms-23-06096-f004:**
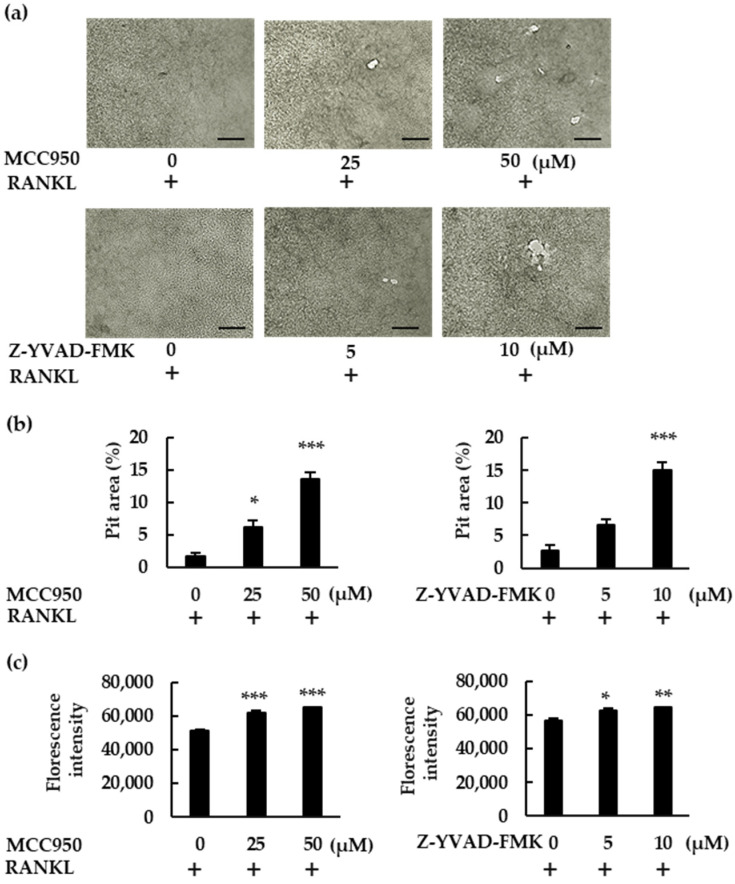
MCC950 and Z-YVAD-FMK upregulate bone-resorption activity in BMMs treated with RANKL. (**a**) BMMs were treated with M-CSF (30 ng/mL) and RANKL (20 ng/mL) in the presence of the indicated concentrations of MCC950 or Z-YVAD-FMK for 4 d. The medium was then replaced with fresh medium containing the same concentrations of these reagents and incubated for another 6 d. After removing the supernatant, we treated the calcium-coated plate with 5% sodium hypochlorite to remove the cells. The plate was then dried and images of the pit-forming areas were captured under a microscope. (**b**) Pit-forming areas were measured using ImageJ software. (**c**) Bone-resorption capacity was also assessed via the fluorescence intensity of the supernatant, using a microplate reader. Scale bars: 150 μm. Results are expressed as the means ± SE of triplicate assays and the images are representative of three independent experiments. * *p* < 0.05, ** *p* < 0.01, *** *p* < 0.001 compared with the control.

**Figure 5 ijms-23-06096-f005:**
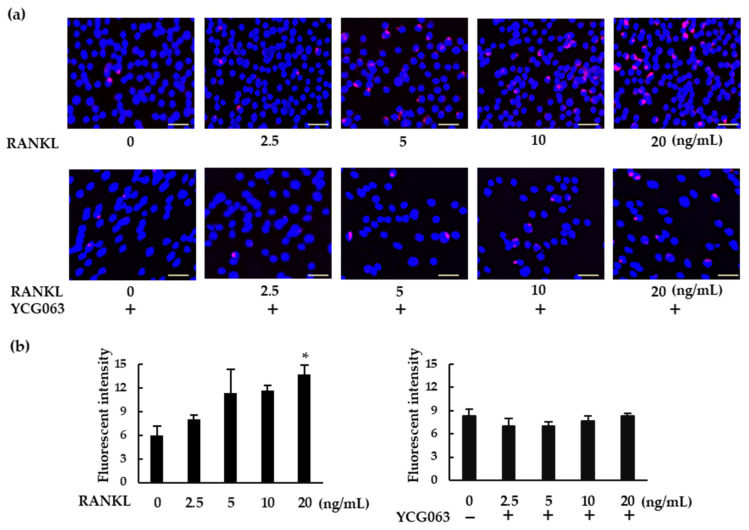
RANKL induces intracellular ROS production in BMMs. (**a**) BMMs were seeded in a 96-well plate and cultured overnight. The cells were then treated with M-CSF (30 ng/mL) and various concentrations of RANKL (0, 2.5, 5, 10, or 20 ng/mL) for 30 min. For the inhibition assays, the cells were pretreated with YCG063 (a ROS inhibitor; 20 μM) in the presence of M-CSF for 1 h. The cells were then stained with a commercial ROS detection kit (red). Hoechst 33342 (blue) was used for counterstaining. ROS-expressing cells were detected using a fluorescence microscope (BZ-X800). (**b**) Fluorescent intensity was measured using a microplate reader. Scale bars: 150 μm. Results are expressed as the means ± SE of triplicate assays and the images are representative of three independent experiments. * *p* < 0.05 compared with the control.

**Figure 6 ijms-23-06096-f006:**
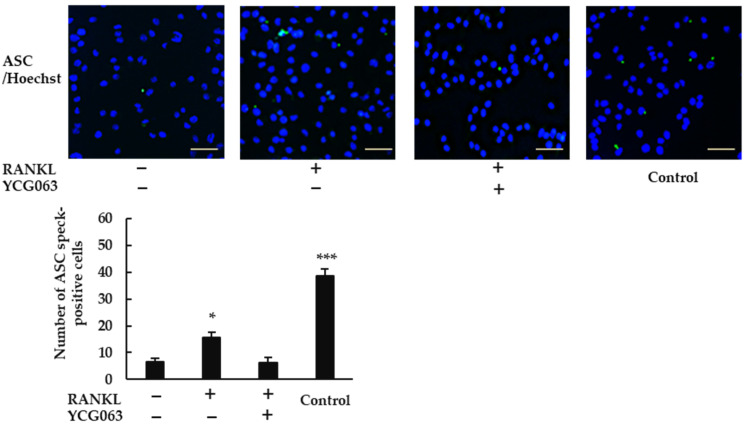
ROS inhibitor YCG063 suppressed RANKL-induced NLRP3 inflammasome assembly. BMMs were treated with M-CSF (30 ng/mL) and RANKL (0 or 20 ng/mL) in the presence or absence of YCG063 (20 μM) for 24 h. For positive controls, cells were primed with LPS (1 μg/mL) for 3 h and further stimulated with ATP (5 mM) for 90 min. The cells were then fixed with 4% paraformaldehyde (PFA) and stained with anti-ASC antibodies (green). Hoechst 33342 (blue) was used for counterstaining. ASC speck formation was then analyzed under a fluorescence microscope (BZ-X800). Scale bars: 200 μm. Results are expressed as the means ± SE of triplicate assays and the images are representative of three independent experiments. * *p* < 0.05, *** *p* < 0.001 compared with the control.

**Figure 7 ijms-23-06096-f007:**
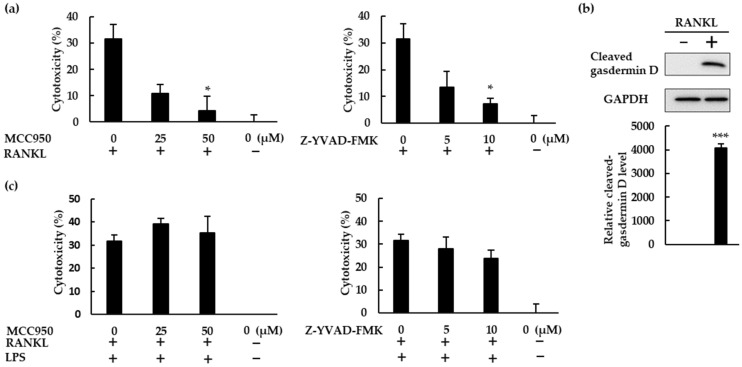
RANKL induced pyroptotic cell death via the NLRP3 inflammasome. (**a**) BMMs were treated with M-CSF (30 ng/mL) and RANKL (10 ng/mL) in the presence or absence of MCC950 or Z-YVAD-FMK for 72 h. (**b**) BMMs were treated with RANKL (0 or 20 ng/mL) and M-CSF (30 ng/mL) for 24 h. Cell lysates were then prepared and subjected to SDS-PAGE followed by Western blotting using anti-gasdermin D and anti-GAPDH antibodies. Relative levels of protein bands were determined using ImageJ software. (**c**) BMMs were treated with RANKL (5 ng/mL) and M-CSF (30 ng/mL) for 48 h, then stimulated with 10 ng/mL LPS in the presence or absence of MCC950 or Z-YVAD-FMK for 24 h. Following incubation, we quantified the level of cytotoxicity by measuring lactate dehydrogenase (LDH) concentrations in the culture supernatant. The results are expressed as the means ± SE of triplicate assays and the images are representative of three independent experiments. * *p* < 0.05, *** *p* < 0.001 compared with the control.

**Figure 8 ijms-23-06096-f008:**
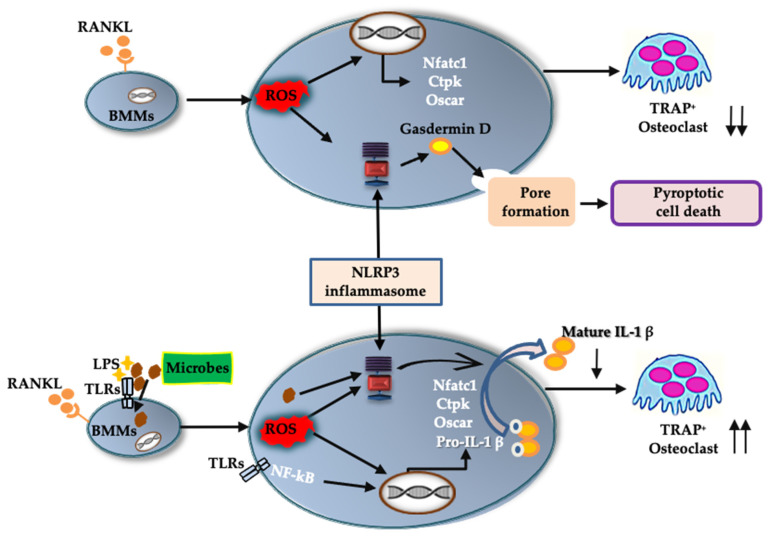
A schematic model of the role of the NLRP3 inflammasome in osteoclast differentiation. In physiological bone remodeling, RANKL activates the NLRP3 inflammasome via ROS production. The activated NLRP3 inflammasome cleaves gasdermin D to its active form. The cleaved gasdermin D induces pyroptotic cell death in RANKL-primed BMMs and downregulates osteoclastic differentiation. In infectious condition, recognition of microbial components by TLRs induces an intracellular signaling to upregulate pro-IL-1β. Microbial components and/or ROS activate NLRP3 inflammasome which processes pro-IL-1β to its mature form. The mature IL-1β promote osteoclastogenic differentiation of RANKL-primed BMMs. ↑↑, acceleration of osteoclast formation; ↓↓, suppression of osteoclast formation.

## Data Availability

The data presented in this study are available on request from the corresponding author.

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
