# Peer review of "NLRP3 Inflammasome Negatively Regulates RANKL-Induced Osteoclastogenesis of Mouse Bone Marrow Macrophages but Positively Regulates It in the Presence of Lipopolysaccharides"

_ijms, 2022, doi:10.3390/ijms23116096_

Round 1

Reviewer 1 Report

In this manuscript, the authors examine the role of the NLRP3 inflammasome in RANKL-induced osteoclastogenesis; NLRP3 inhibitors promoted osteoclast formation from bone marrow derived macrophages (BMM) by RANKL. On the other hand, in the presence of LPS, NLRP3 inhibitors inhibited osteoclastogenesis by RANKL. Treatment with RANKL and LPS enhanced IL-1 production in RANKL-induced osteoclast formation, and assays using IL-RA suggested the involvement of IL-1 in RANKL plus LPS-induced osteoclastogenesis. On the other hand, ROS activation was observed in RANKL-induced osteoclast formation, and experiments using ROS inhibitors revealed that ROS activation activated the NLRP3 inflammasome and induced pyroptosis. This is an interesting and significant study that reveals the function of the NLRP3 inflammasome in RANKL-induced osteoclastogenesis. However, there are some shortcomings in the paper, and we believe that additional experiments are needed.

Major points

  1. The effect of NLRP3 inflammasome on RANKL-induced osteoclast formation is evaluated only on the number of osteoclasts. Since osteoclast formation occurs with activation of osteoclast-related genes, mRNA expression of NFATc1, OSCAR, CTSK, etc. should also be examined.
  2. The gene expressions of NLRP3 inflammasome during osteoclastogenesis were measured by real-time PCR alone, but it is not sufficient to measure the gene expression. The innflammasome-associated protein expression should also be verified by Western blotting.
  3. It is not appropriate to assume that pyroptosis occurs in RANKL-induced osteoclastogenesis by measuring only LDH. In addition to these data, immunostaining for Caspase1 or Gasdermin D should be performed.

4.Throughout the paper, the system of experiments performed is rather complex, so a summary schema at the end of the paper(as Figure 7) would make it easier for the readers to understand.

Minor points

Figure 1

(c) and (d) should be integrated into the same graph.

Figure 2

In (a) and (b), it should be noted what the ‘control’ in the graphs indicate.

Figure 3

The number of osteoclasts formed by RANKL alone in (a) and (b) should be comparable. However, in (a), the number of osteoclasts is about 160, and in (b), the number of osteoclasts is about 80. The osteoclast formation assay in (b) is probably not working well. This assay in (b) should be performed again.

Figure 4

(a) It should be noted what the blue and red colors in the photographs indicate, respectively.

Figure 6

It should be noted what the blue and green color in the photographs indicate, respectively. Furthermore, it should be noted what the ‘Control’ in the rightmost photograph indicates. The expressions of ASC are very low in these photos, but is pyroptosis really occurring?

Author Response

Reply to Reviewer 1:

(Manuscript ID: ijms-1684310)

Thank you for giving us the important comments and suggestion.

We have addressed your comments as follows:

Major Points:

  1. The effect of NLRP3 inflammasome on RANKL-induced osteoclast formation is evaluated only on the number of osteoclasts. Since osteoclast formation occurs with activation of osteoclast-related genes, mRNA expression of NFATc1, OSCAR, CTSK, etc. should also be examined.

We have performed qRT-PCR to measure the expression of Nfatc1, Oscar, and Ctsk genes in the presence of inflammasome inhibitors. The results are shown in the new "Figure 3e".

  1. The gene expressions of NLRP3 inflammasome during osteoclastogenesis were measured by real-time PCR alone, but it is not sufficient to measure the gene expression. The inflammasome-associated protein expression should also be verified by Western blotting.

According to the suggestion, we have performed the Western blot analysis to examine the inflammasome-associated protein expression in RANKL-treated BMMs. The results are shown in the new "Figure 1d".

  1. It is not appropriate to assume that pyroptosis occurs in RANKL-induced osteoclastogenesis by measuring only LDH. In addition to these data, immunostaining for Caspase1 or Gasdermin D should be performed.

According to the suggestion, we have performed the immunostaining for caspase1 and gasdermin D. The result of the immunostaining for caspase1is shown in the new "Figure 1d". The result of the immunostaining for gasdermin D is shown in the new 'Figure 7b".

  1. Throughout the paper, the system of experiments performed is rather complex, so a summary schema at the end of the paper (as Figure 7) would make it easier for the readers to understand.

According to the suggestion, we have added the summary schema to the new "Figure 8".

Minor Points:

  1. Figure 1: (c) and (d) should be integrated into the same graph.

According to the suggestion, we have combined the Figure 1c and 1d into the new Figure 1c.

  1. Figure 2: In (a) and (b), it should be noted what the ‘control’ in the graphs indicate.

We have added the explanation of the ‘control’ to the figure legend.

  1. Figure 3: The number of osteoclasts formed by RANKL alone in (a) and (b) should be comparable. However, in (a), the number of osteoclasts is about 160, and in (b), the number of osteoclasts is about 80. The osteoclast formation assay in (b) is probably not working well. This assay in (b) should be performed again.

According to the suggestion, we have replaced the graph in figure 3 (b) with the new result.

  1. Figure 4: (a) It should be noted what the blue and red colors in the photographs indicate, respectively.

We have added the explanation of the blue and red colors to the figure legends.

  1. Figure 6: It should be noted what the blue and green color in the photographs indicate, respectively. Furthermore, it should be noted what the ‘Control’ in the rightmost photograph indicates. The expressions of ASC are very low in these photos, but is pyroptosis really occurring?

We have added the explanation of the blue and green colors and the explanation of the ‘control’ to the figure legends. The expressions of ASC were significantly different as shown in the graph of Figure 6.

Reviewer 2 Report

This study investigated the mechanism underlining the NLRP3 inflammasome regulation of osteoclastogenesis of mouse bone marrow macrophages. The rational behind the experiment was clear and straight forward. The manuscript is almost well written.

While many different sources are used to set up the study in the introduction, little previous evidence is stated. The introduction is thus short and poorly sets up the rationale for the study. More attention to how this study fits into previous work

There are some minor grammar issues that should be fixed in order to aid the accessibility of the results to the reader.

Author Response

Reply to Reviewer #2:

(Manuscript ID: ijms-1684310)

Thank you for giving us the important comments and suggestion.

We have addressed your comments as follows:

  1. While many different sources are used to set up the study in the introduction, little previous evidence is stated. The introduction is thus short and poorly sets up the rationale for the study. More attention to how this study fits into previous work

According to the suggestion, we have modified the Introduction.

  1. There are some minor grammar issues that should be fixed in order to aid the accessibility of the results to the reader.

According to the suggestion, we have revised our manuscript.

Round 2

Reviewer 1 Report

The revised manuscript has been written more clearly than the former one with the additional experiments I suggested.